# Impact of Ovarian Endometrioma and Surgery on Reproductive Outcomes: A Single-Center Spanish Cohort Study

**DOI:** 10.3390/biomedicines11030844

**Published:** 2023-03-10

**Authors:** Alicia Hernández, Angela Sanz, Emanuela Spagnolo, Ana Lopez, Paloma Martínez Jorge, Silvia Iniesta, Elena Rodríguez, Sara Fernández Prada, David Ramiro-Cortijo

**Affiliations:** 1Department of Obstetrics and Gynecology, Hospital Universitario La Paz, 28046 Madrid, Spain; 2Instituto de Investigación IdiPaz, Hospital Universitario La Paz, 28029 Madrid, Spain; 3Department of Obstetrics and Gynaecology, Faculty of Medicine, Universidad Autónoma de Madrid, 28029 Madrid, Spain; 4Department of Physiology, Faculty of Medicine, Universidad Autónoma de Madrid, 28029 Madrid, Spain; 5Instituto Universitario de Estudios de la Mujer (IUEM), Universidad Autónoma de Madrid, 28049 Madrid, Spain

**Keywords:** endometriosis, in vitro fertilization, surgically treatment, ovarian endometrioma, ovarian reserve, reproductive outcomes

## Abstract

Anti-Müllerian hormone (AMH) and antral follicular count (AFC) decrease in women with ovarian endometrioma (OMA) and in vitro fertilization (IVF). In addition, these parameters drop even further when women with OMA undergo surgery. In this study, the primary aim was to compare the reproductive variables in IVF-treated women with and without endometriosis. The secondary aim was to explore if the reproductive variables were modified by endometrioma surgery. In this retrospective study, 244 women undergoing IVF were enrolled at the Hospital Universitario La Paz (Madrid, Spain). Women were categorized as OMA not surgically treated (OMA; *n* = 124), OMA with surgery (OMA + S; *n* = 55), and women with infertility issues not related to OMA (control; *n* = 65). Demographic and clinical variables, including age, body mass index (BMI), and reproductive (AMH, AFC, number of extracted oocytes, and transferred embryos) and obstetrical data (biochemical pregnancy and fetal heart rate at 6 weeks) were collected. Adjusted logistic regression models were built to evaluate reproductive and pregnancy outcomes. The models showed that women with OMA (with and without surgery) had significantly decreased levels of AMH and AFC and numbers of cycles and C + D embryos. Women with OMA + S had similar rates of pregnancy to women in the control group. However, women with OMA had lower biochemical pregnancy than controls (aOR = 0.08 [0.01; 0.50]; *p*-value = 0.025). OMA surgery seems to improve pregnancy outcomes, at least until 6 weeks of gestation. However, it is important to counsel the patients about surgery expectations due to the fact that endometrioma itself reduces the quality of oocytes.

## 1. Introduction

Endometriosis is a chronic gynecological disease characterized by the presence of functionally active endometrial tissue (glandular epithelium and stroma) outside the uterine cavity, which induces a chronic inflammatory reaction [1]. The prevalence was estimated at around 10–15% of women of reproductive age. The main clinical complications are pelvic pain and infertility issues [2]. The infertility rate in women affected by endometriosis is estimated to be between 30% and 50% [3]. Ovarian endometriosis (OMA) is present in 17–44% of women with endometriosis. There are data indicating that OMA is not presumed to have an impact on ovulation, and there are concerns about its adverse effects on ovarian reserve [4]. Thus, the association between OMA and decreased ovarian reserve has been extensively established [5,6,7,8]. However, OMA usually coexists with deep endometriosis, among other manifestations. For this reason, the impact of OMA isolated on female fertility is presumed to be overestimated and needs to be elucidated. The current therapeutic option for endometriosis depends on the clinical presentation and pain symptomatology. When pain is the main issue and the women have no contraindications, the international gynecological societies recommend starting with pharmacological treatment, which includes a wide possibility, such as gonadotropin-releasing hormone analogs and contraceptives [9,10,11]. However, when it comes to infertility, pharmacological treatment is inefficient [12]. The non-pharmacological options include ultrasound-guided aspiration, ablative techniques, and surgical resection. It is preferable to use excision or ablation of the cysts rather than drainage [13].

Ovarian reserve is defined as the number and quality of follicles in the ovary at any set time [14]. Clinically, the quality of follicles can be evaluated with validated biomarkers of ovarian reserve. Generally, quantitative tools such as antral follicle count (AFC) and serum anti-Müllerian hormone (AMH) levels are widely used. The number of oocytes retrieved (NOR) during in vitro fertilization (IVF) cycles can also be considered. The most reliable and extensively used biomarker has been the level of AMH, especially when using assisted reproductive technology, such as IVF, due to its consistency throughout the menstrual cycle and its correlation with age, AFC, and the response to controlled ovarian hyperstimulation [15,16].

In women undergoing IVF cycles, there is solid evidence showing that OMA leads to a reduced NOR and a higher risk of poor ovarian response to ovarian stimulation cycles [17,18]. Moreover, in the last few years, increasing emphasis has been placed on the surgical management of OMA to improve the reproductive opportunities of these women [19]. However, this issue is controversial; while some authors suggested that the lower NOR in patients with OMA occurs secondarily to endometrioma surgery [20], others support that the endometrioma itself has a negative impact on ovarian response to ovarian stimulation [21]. However, regarding the pregnancy outcomes after IVF cycles, pregnancy seems to not be affected in women after endometrioma surgery [17]. Thus, the negative effect of endometrioma on the ovarian reserve of women is unclear. Moreover, there are poor data clarifying whether surgical treatment of women with OMA could modify the successful use of IVF cycles. This study has the primary outcome of comparing the reproductive variables in women undergoing IVF with and without endometriosis; the secondary outcome is to explore if the reproductive variables were modified by ovarian endometrioma surgery.

## 2. Materials and Methods

### 2.1. Study Design

This was an observational, non-interventional, and retrospective study that enrolled 275 women undergoing IVF cycles and OMA diagnoses followed in the Obstetrics and Gynecology Service at the Hospital Universitario La Paz (HULP; Madrid, Spain) between January 2018 and December 2021. The inclusion criteria were women aged between 18 and 40 years and receiving IVF cycles. The exclusion criteria applied were women in menopausal stage, women with deep endometriosis, coexistence of other adnexal masses (teratoma, tubo-ovarian abscess), previous ovarian surgery, and missing clinical data. The women were categorized as OMA not surgically treated (OMA; *n* = 124), OMA with surgery (OMA + S; *n* = 55), and women with infertility issues not related to OMA (control; *n* = 65; Figure 1). The group of women with OMA + S received IVF after laparoscopic cystectomy.

This study was performed in accordance with the Declaration of Helsinki regarding studies in humans and was approved by the HULP Research Ethics Committee (Ref. 07/207514.9/22; HULP code PI-5095). The confidentiality and anonymity of the data were guaranteed at every moment of the study protocol.

### 2.2. Biochemical, Reproductive, and Obstetrical Variables

Before starting ovarian stimulation cycles, a blood sample was obtained by venipuncture in anticoagulant tubes to determine AMH levels using the automated electrochemiluminescence immunoassay kit (Elecsys analyzer, 1.0–2.6% CV, 0.046–20.8 ng/mL Roche Diagnostics, Barcelona, Spain). In OMA + S group, AMH levels were analyzed 6 months after surgery. The AFC was performed by transvaginal ultrasound (Mindray DC60 ultrasound scanner, Olympus, Tokyo, Japan). The follicles <10 mm were counted according to the standardized technique described by Broekmans [22]. AMH and AFC were considered ovarian reserve markers.

Data for the demographic and clinical variables, including IVF parameters (number of IVF cycles performed, number of >16 mm follicles, number of AB type embryos obtained, number of CD type embryos obtained, type of transfer (no transfer, fresh transfer, or cryotransfer), and biochemical pregnancy (β-chorionic hormone, βCH, positive or negative)), were collected from medical records. The plasma βCH was determined within 12–14 days after transfer, whether fresh or delayed transfer, using an automated electrochemiluminescence immunoassay kit (Elecsys free βhCG, ≥20% CV, 0.3–190 mIU/mL Roche Diagnostics, Barcelona, Spain). The embryo transfers were performed with a soft echo-guided transfer catheter (Labotec, Leon, Spain).

The women with more than 2 missing values in data from the medical records were considered as significantly missing data and were also excluded from the analysis.

### 2.3. Statistical Analysis

Data analysis was performed using R software (version 4.2.2, R Core Team 2022, Vienna, Austria) in RStudio (version 2022.07.2 + 576, RStudio, PBC, 2009–2022, Inc., Vienna, Austria) using the *rcompanion*, *dplyr*, *tidyverse*, *devtools*, *arsenal*, *compareGroups*, *rio*, and *oddsratio* packages.

The data were expressed as the median and interquartile range [Q1; Q3] for quantitative variables. The sample size (*n*) and relative frequency (%) were used to describe qualitative variables. The univariate analysis was performed by Kruskal–Wallis test followed by the Dunnett post hoc test or Mann–Whitney U test depending on the comparison groups. Chi-squared corrected by Fischer’s exact test was used to compare proportions.

To evaluate the effect of endometriosis (with and without surgery) in women undergoing IVF on the reproductive variables, linear generalized regression models (LRMs) were built considering women without endometriosis (control group) as the reference. Similarly, logistic regression models (LgRMs) were built to evaluate the pregnancy outcomes. Models were adjusted by age, smoking habits, body mass index, and other significant variables in the univariate analysis. The coefficient (β) ± standard error (SE) was extracted from the LgRM, and the OR and 95% confidence interval ([95% CI]) were extracted from the LgRM. The *p*-value was extracted from each association factor. In this study, techniques for imputing data were not used. A *p*-value < 0.05 was considered to indicate a significant difference.

## 3. Results

### 3.1. Characteristics of the Cohort

In the cohort, the median age was 35.0 (Min = 22.0; Max = 40.0) years old. No statistical differences were detected in age, weight, height, and BMI between the analyzed groups. In addition, there was a significantly lower prevalence of Caucasian women in the control group than in the endometriosis groups (Table 1).

The endometriosis diagnoses represented 71.6% (163/229) of the cohort, with 26.4% (43/163) being surgically treated. The surgical intervention was unilateral cystectomy in 16.9% and bilateral cystectomy in 8.1%. In addition, 1.9% were unilateral adnexectomies. Of the surgeries for OMA, 65.4% were for clinical symptomatology, followed by 19.2% for suspicious ultrasound features, and 15.4% for fertility issues. Endometriosis was unilateral in 70.2% of women without surgery. No statistical association was detected between endometriosis laterality and treatment by surgery (χ^2^ < 0.001; *p*-value > 0.99). The endometrioma size was significantly larger in women who had surgical treatment than in women without surgery (OMA = 2.5 [1.8; 3.1] cm, OMA + S = 5.3 [3.5; 6.8] cm; *p*-value < 0.001). In addition, the rate of women with endometrioma > 4 cm was higher in OMA + S than in OMA women (χ^2^ = 19.02; *p*-value < 0.001).

The neutrophil level was significantly higher in women with OMA (5.19 [3.73; 6.04] 103/μL) than in controls (3.82 [3.04; 5.11] 103/μL) and women with OMA + S (3.96 [2.95; 5.52] 103/μL).

### 3.2. Reproductive Characteristics in Women Undergoing IVF

In terms of reproductive parameters, women with endometriosis exhibited significantly reduced levels of anti-Müllerian hormone, follicular count, number of cycles, number of oocytes in each cycle, and number of category C and D embryos per cycle. However, these parameters did not show significant differences between women with OMA and women with OMA + S (Table 2).

### 3.3. Pregnancy Outcomes in Women Undergoing IVF

No twin pregnancies were detected in any of the study groups. Although β-chorionic hormone plasma levels did not differ between groups, the percentage of women with a biochemically positive pregnancy was significantly different between groups. Women with OMA + S had higher rates of biochemical pregnancy than women with OMA. Women in the OMA group had a reduced prevalence of positive fetal heart rate compared to controls. From 12 weeks of gestation, the fetal loss ratios were similar in all groups (Table 3).

### 3.4. Endometriosis Association Factors with Reproductive Variables and Pregnancy Outcomes

Considering the women in the control group as a reference, our adjusted models showed that women with OMA (with and without surgery) had significantly decreased levels of anti-Müllerian hormone, follicular count, number of cycles, and number of embryos of category C + D transferred per cycle. In addition, only in women with OMA + S, the number of oocytes per cycle was also reduced (Table 4). However, women with OMA + S did not show different risks in pregnancy outcomes compared to women in the control group. Nevertheless, the women with OMA had an 0.08 [0.01; 0.50] risk of not having a fetal heart rate at 6 weeks of gestation (*p*-value = 0.025; Table 4).

A secondary analysis excluding the control women group and using the women with OMA as the reference demonstrated that the levels of anti-Müllerian hormone were significantly reduced in women with OMA + S (β = −0.66 ± 0.28; *p*-value = 0.020). However, the risks of the rest of the reproductive and pregnancy outcomes were not modified by the surgical treatment of endometriosis in women undergoing IVF (Table 5).

## 4. Discussion

Overall, endometriosis decreases the ovarian reserve [23], and considering the reproductive outcomes of the women, the role of OMA surgery is still controversial, particularly in managing fertility issues where the used therapeutic protocols have side effects. From a conservative point of view, leaving the OMA lesions may decrease the ovarian reserve and the odds of pregnancy. Moreover, this strategy can lead to some complications, such as cyst rupture or endometriosis-associated malignancy. On the other hand, surgical treatment may not only damage the ovarian response due to the removal of healthy ovarian tissue, but also produce local inflammation and vascular compromise following surgical coagulation [24]. Regarding our data for women with IVF, low levels of reproductive variables (AMH, follicle count, number of oocytes, and number of embryos) were observed in women with OMA compared to women in the control group (infertility issue not related to OMA). Furthermore, reproductive and pregnancy outcomes were not modified by the OMA surgical treatment. Finally, women with surgically treated OMA, compared to women with OMA without surgery, showed a high rate of fetal heart rate at 6 weeks of pregnancy. These observations can be key due to obstetrical results in young patients with endometriosis undergoing surgery.

Previous studies have shown a quicker decline in AMH level and AFC in women with OMA compared to age- and BMI-matched counterparts without endometriosis [3,5,8,18,25,26]. In our study, the adjusted models showed similar results, with a consistent and significant decrease in AMH levels in women with endometriosis and surgical treatment (β = −0.66 ± 0.28). In the literature, this fact was postulated to be due to the distortion in the ovarian anatomy and inflammatory environment, and thus, endometriomas might be a detrimental factor for fertility [27]. Other studies showed follicle density was reduced in the cortex surrounding the endometrioma, suggesting that the endometrioma itself damaged the ovary [28].

Regarding the IVF outcomes, patients with OMA showed significantly low NOR, especially in the first cycle, in comparison to the control group. A low number of embryos with C + D categories was also shown in patients with OMA. However, the number of embryos with A + B categories was similar in women with or without endometrioma. These results were consistent with the data of other authors [21,26,29] and were mainly attributed to the low number of available embryos in patients with OMA. These findings could indicate that IVF has benefits in patients with OMA.

Regarding the effect of surgery on ovarian reserve, our study did not show any significant differences in AMH levels and AFC between women with or without OMA surgery. Other studies have shown a significant decrease in ovarian reserve markers after surgery [3,8,21,30,31]. In contrast, the meta-analysis of Muzi et al. reported no significant change in AFC after surgery for ovarian endometriomas [32]. As the authors explain, these controversial results can be attributed to the destruction of healthy ovarian tissue, which cannot be monitored and should be an aspect to further consider. Additionally, in women with OMA, the AMH levels and AFC reduction are also influenced by surgical technique [33] and the surgeon’s expertise [34]. Some authors argue that a decrease in the levels of AMH is recoverable after 12 months [35], while others argue that it is irreversible [31]. In addition, when analyzing the histology of OMA cystectomy on ovarian reserve, some authors have shown that the ovarian tissue removed along with endometriomas was mostly fibrotic or non-functional [34]. This is interesting due to the reproductive implications for patients and for correctly evaluating the fertilization treatment. However, prospective multicenter studies on large samples would be necessary to draw definitive conclusions. Moreover, the Raffi et al. systematic review found a 40% fall in AMH plasma concentration after ovarian cystectomy for endometriomas [30]. However, it is necessary to consider the high heterogeneity included in the review. Only two of the studies evaluated the AMH level 6–9 months after surgery. In the other four studies, the median size of the cyst was not specified [30]. These observations could be important but need to be prudently evaluated. Furthermore, in all studies, the sample size enrolled was low. Considering our data where the recruitment effort was related to 164 women with OMA, a significant reduction in AMH, follicular count, and oocytes per cycle was shown. Interestingly, regarding embryos/cycle, only embryo C + D numbers were significantly lower in patients surgically treated for OMA compared to OMA patients without surgery. This result confirms that endometrioma itself reduces the quality of oocytes. This finding is paramount in patient counseling. Table 6 summarizes some studies exploring the treatment of women with OMA before IVF techniques and its effect on reproductive and pregnancy outcomes.

It seems evident that larger lesion size, presence of bilaterality, histological damage, and lower ovarian reserve affect the reproductive outcomes [23,41,42]. In our study, bilateral cystectomy did not present a risk factor for reproductive and pregnancy variables. On the contrary, in women without surgery, bilateral OMA was associated with a lower number of embryos categorized as A + B compared to unilateral OMA. This result can be explained by the important role of endometrioma in reducing oocyte quality.

On the other hand, the pregnancy outcome needs further attention. It was observed that patients with OMA (with or without surgery) developed lower fetal heart rates at 6 and 12 weeks of gestation compared to controls. However, the differences in miscarriage and abortion were not significant between groups. Recently, a meta-analysis has shown significantly low clinical pregnancy in women with or without OMA undergoing IVF, but no difference in live birth rate [29]. Wu et al. performed an observational study in women undergoing IVF and showed that clinical pregnancy and implantation rates were similar in women with endometriomas compared to women without endometrioma [26]. In our study, the prevalence of biochemical pregnancy after IVF in women with endometrioma was low compared to that in women with surgically treated endometriosis (43.6% versus 55.3%). Women with OMA + S and control showed similar pregnancy rates at 6 weeks (92.6% versus 95.5%). On the other hand, when OMA was considered the reference, we showed that OMA + S did not change its association. This result could indicate that surgery for OMA would improve the pregnancy rate. We could explain this result by highlighting the pathogenesis of endometriosis. In the cyst, the endometriotic cells produce high concentrations of iron, which can produce reactive oxygen species (ROS), blocking embryo implantation [43]. Other studies showed that biochemical and clinical pregnancy and live birth rate were not associated with the excision of endometrioma [26,38]. In contrast, Barri et al. evaluated 825 women with endometriosis-related infertility and showed that pregnancy rates were significantly higher in women with surgically treated OMA and subsequent IVF than in those with surgery alone, IVF alone, or no treatment [44]. A systematic review suggested that the implantation and clinical pregnancy were reduced in women with concomitant and severe endometriosis [45]. Other authors showed higher rates of clinical pregnancy and implantation in women who underwent endometriosis surgery than in those who had endometriosis that was not surgically treated [39]. Considering the literature and our data on women with endometriosis, it appears that the pregnancy outcomes of IVF-mediated gestations are improved if the endometriosis condition is previously treated surgically, with this surgery being safe for gestational viability. Additionally, counseling regarding ovarian reserve and surgery in women with endometriosis and reproductive desire is a cornerstone to implement in the guidelines.

### Strengths and Limitations

This study adds knowledge to the available evidence regarding the influence of endometrioma and its surgical management on ovarian reserve and IVF reproductive outcomes. As a strength, there are few studies that include IVF outcomes and compare women without and with endometriosis (with and without surgery) simultaneously. In endometriosis patients, the importance of evaluating reproductive results is due to women’s concerns about surgery and any ovarian reserve reduction [45], particularly in the case of reproductive desire. In this study, a multiple regression analysis adjusting for age, smoking habits, race, body mass index, and neutrophil levels was performed. This analysis indicated that there was no difference in ovarian reserve between groups.

The present study is not free of limitations. The limitations include the lack of a long-term postoperative follow-up to evaluate ovarian function recovery and the retrospective design, in which it is difficult to establish causality. However, the study design is important in the research context to set preliminary factors for new guidelines and protocols. Nevertheless, it would be necessary for longitudinal and prospective studies to reinforce the conclusion in women with endometriosis undergoing IVF. A further limitation is the lack of evaluation of factors that may affect ovarian reserve, such as bipolar electrocoagulation. Indeed, other findings showed the impact of surgical techniques on reproductive outcomes [20]. It will be interesting to deeply analyze these data in future studies.

## 5. Conclusions

Women with OMA (with/without surgery) could present a proinflammatory environment that may compromise the viability of the IVF technique and subsequent gestation due to a reduction in ovarian reserve markers. In these women, the number of C + D embryos was significantly lower than that in the control group. Regarding the pregnancy outcome, laparoscopic cystectomy (OMA surgery) seems to improve pregnancy outcomes, at least until 6 weeks of gestation. However, it is important to adequately select the women who may benefit from surgery. Indeed, a global change in the clinical approach to endometriosis, focusing on pain, fertility, and risk factors of complications, is recommendable. Finally, it is important to counsel the patients about surgery expectations because endometriosis itself reduces the quality of oocytes.

## Figures and Tables

**Figure 1 biomedicines-11-00844-f001:**
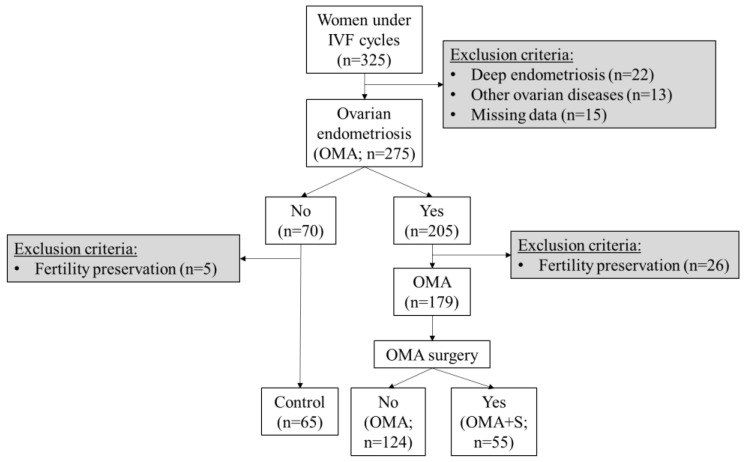
Study design and flow chart. Sample size (n) is indicated in each category. The control group comprised 36 women with tubal factor, 15 women with low ovarian reserve, and 14 women with polycystic ovary syndrome.

**Table 1 biomedicines-11-00844-t001:** Sociodemographic and biochemical variables between women undergoing IVF with and without endometriosis.

	Control (*n* = 65)	OMA (*n* = 121)	OMA + S (*n* = 43)	*p*-Value
Age (years)	35.0 [33.0; 37.0]	35.0 [33.0; 38.0]	35.0 [32.5; 37.0]	0.33
Smoking habits	14.5% (10)	6.8% (8)	10.0% (4)	0.24
Race				
Caucasian	89.8% (53)	99.0% (97)	97.4% (37)	0.019
Non-Caucasian	10.2% (10)	1.0% (1)	2.6% (1)	
Categorized				
<25 years	3.1% (2)	0.0% (0)	0.0% (0)	0.56
25–29 years	9.2% (6)	8.3% (10)	4.7% (2)	
30–34 years	40.0% (26)	43.0% (52)	48.8% (21)	
35–40 years	47.7% (31)	48.8% (59)	46.5% (20)	
Weight (kg)	66.5 [55.0; 74.2]	62.0 [56.0; 67.0]	62.0 [55.5; 72.0]	0.26
Height (m)	1.65 [1.60; 1.69]	1.65 [1.60; 1.69]	1.65 [1.61; 1.70]	0.82
BMI (kg/m^2^)	23.8 [21.8; 26.7]	22.8 [20.7; 25.2]	23.0 [20.9; 25.8]	0.27
Leucocytes (10^3^/μL)	6.34 [5.63; 7.89]	6.60 [5.46; 7.82]	6.16 [5.12; 7.42]	0.34
Neutrophils (10^3^/μL)	3.82 [3.04; 5.11] ^a^	5.19 [3.73; 6.04] ^b^	3.96 [2.95; 5.52] ^a^	0.001

For the quantitative variables, data show median and interquartile range [Q1; Q3], and the *p*-value was extracted from the Kruskal–Wallis test. Different letters indicate significant differences (*p*-value < 0.05) as determined by Dunnett post hoc test. For qualitative variables, data show the relative frequency (%) and sample size (*n*), and the *p*-value was extracted from the Fisher exact test. Ovarian endometriosis (OMA); surgery (S); body mass index (BMI).

**Table 2 biomedicines-11-00844-t002:** Reproductive variables between women undergoing IVF with and without endometriosis.

	Control (*n* = 65)	OMA (*n* = 121)	OMA + S (*n* = 43)	*p*-Value
AMH (ng/mL)	3.10 [1.55; 4.70] ^a^	1.70 [0.87; 3.32] ^b^	1.20 [0.78; 1.97] ^b^	<0.001
Follicular count	12.5 [9.0; 19.2] ^a^	7.0 [4.0; 10.0] ^b^	7.0 [5.8; 10.0] ^b^	<0.001
Number of cycles	2.0 [1.0; 3.0] ^a^	1.0 [1.0; 2.0] ^b^	1.0 [1.0; 2.0] ^b^	0.001
Number of follicles (>16 mm)				
1st Cycle	9.0 [3.0; 12.0]	8.0 [4.0; 10.0]	7.0 [3.0; 9.5]	0.25
2nd Cycle	0.0 [0.0; 11.0]	0.0 [0.0; 7.0]	0.0 [0.0; 5.5]	0.33
3rd Cycle	0.0 [0.0; 0.0]	0.0 [0.0; 0.0]	0.0 [0.0; 1.0]	0.17
Follicles/cycle	8.7 [6.0; 13.5]	9.0 [5.0; 12.0]	8.0 [3.50; 11.5]	0.68
Number of oocytes				
1st Cycle	9.0 [6.0; 14.0] ^a^	6.0 [4.0; 9.0] ^b^	5.0 [3.0; 8.0] ^b^	<0.001
2nd Cycle	4.0 [0.0; 11.0] ^a^	1.0 [0.0; 8.0] ^b^	0.0 [0.0;4.0] ^b^	0.021
3rd Cycle	0.0 [0.0; 5.0] ^a^	0.0 [0.0; 0.0] ^b^	0.0 [0.0; 1.5] ^a,b^	0.024
Oocytes/cycle	9.0 [6.0; 15.0] ^a^	7.0 [5.0; 11.3] ^b^	6.0 [3.0; 12.5] ^b^	0.014
1st Embryo transfer				
No Transfer	32.3% (21)	41.3% (50)	30.2% (13)	0.19
Fresh	23.1% (15)	10.7% (13)	18.6% (8)	
Cryotransfer	44.6% (29)	47.9% (58)	51.2% (22)	
2nd Embryo transfer				
No Transfer	22.5% (9)	38.1% (24)	45.5% (10)	0.37
Fresh	17.5% (7)	14.3% (9)	13.6% (3)	
Cryotransfer	60.0% (24)	47.6% (30)	40.9% (9)	
3rd Embryo transfer				
No Transfer	20.8% (5)	27.3% (6)	41.7% (5)	0.28
Fresh	8.3% (2)	18.2% (4)	25.0% (3)	
Cryotransfer	70.8% (17)	54.5% (12)	33.3% (4)	
Number of embryos (A + B category)				
1st Cycle	0.0 [0.0; 1.0]	0.0 [0.0; 1.0]	0.0 [0.0; 1.0]	0.89
2nd Cycle	0.0 [0.0; 1.0]	0.0 [0.0; 0.0]	0.0 [0.0; 0.0]	0.16
3rd Cycle	0.0 [0.0; 0.0]	0.0 [0.0; 0.0]	0.0 [0.0; 0.0]	0.58
Embryos A + B/cycle	0.3 [0.0; 1.0]	0.5 [0.0; 1.0]	0.3 [0.0; 1.0]	0.92
Number of embryos (C + D category)				
1st Cycle	3.0 [1.0; 7.0] ^a^	1.0 [0.0; 3.0] ^b^	1.0 [0.0; 3.0] ^b^	<0.001
2nd Cycle	1.0 [0.0; 4.0] ^a^	0.0 [0.0; 2.0] ^b^	0.0 [0.0; 1.0] ^b^	0.001
3rd Cycle	0.0 [0.0; 1.0] ^a^	0.0 [0.0; 0.0] ^b^	0.0 [0.0; 0.0] ^a,b^	0.009
Embryos C + D/cycle	4.0 [2.7; 7.0] ^a^	2.0 [1.0; 3.5] ^b^	2.0 [0.4; 3.0] ^b^	<0.001

For the quantitative variables, data show median and interquartile range [Q1; Q3], and the *p*-value was extracted from the Kruskal–Wallis test. Different letters indicate significant differences (*p*-value < 0.05) as determined by Dunnett post hoc test. For qualitative variables, data show the relative frequency (%) and sample size (*n*), and the *p*-value was extracted from the Fisher exact test. Ovarian endometriosis (OMA); surgery (S); anti-Müllerian hormone (AMH).

**Table 3 biomedicines-11-00844-t003:** Pregnancy outcomes between women undergoing IVF with and without endometriosis.

	Control (*n* = 65)	OMA (*n* = 121)	OMA + S (*n* = 43)	*p*-Value
βCH (mIU/mL)	590 [42.0; 1410]	668 [172.0; 1764]	860 [358.0; 1616]	0.40
Biochemical pregnancy (Positive βCH)	63.1% (37)	43.6% (44)	55.3% (21)	0.044
Positive FHR at 6 weeks	95.5% (42)	79.4% (54)	92.6% (25)	0.034
Positive FHR at 12 weeks	97.0% (32)	80.0% (44)	88.9% (16)	0.05
Miscarriage (≤12 weeks)	13.5% (5)	22.4% (13)	5.3% (1)	0.22
Abortion (>12 weeks)	0.0% (0)	13.6% (6)	11.8% (2)	>0.99

For the quantitative variables, data show median and interquartile range [Q1; Q3], and the *p*-value was extracted from the Kruskal–Wallis test. For qualitative variables, data show the relative frequency (%) and sample size (*n*), and the *p*-value was extracted from the Fisher exact test. Ovarian endometriosis (OMA); surgery (S); β-chorionic hormone (βCH); fetal heart rate (FHR).

**Table 4 biomedicines-11-00844-t004:** Associations of endometriosis with reproductive and pregnancy outcomes.

**Reproductive Variables**	**OMA**	** *p* ** **-Value**	**OMA + S**	** *p* ** **-Value**
AMH (ng/mL)	−1.61 ± 0.40	<0.001	−2.34 ± 0.47	<0.001
Follicular count	−7.20 ± 1.11	<0.001	−7.51 ± 1.33	<0.001
Number of cycles	−0.68 ± 0.14	<0.001	−0.65 ± 0.16	<0.001
Oocytes/cycle	−1.86 ± 1.41	0.19	−3.43 ± 1.66	0.040
Embryos C + D/cycle	−2.49 ± 0.72	0.001	−2.26 ± 0.84	<0.001
**Pregnancy Outcomes**	**OMA**	** *p* ** **-Value**	**OMA + S**	** *p* ** **-Value**
Biochemical pregnancy (Positive βCH)	2.19 [0.94; 5.24]	0.07	2.22 [0.81; 6.24]	0.12
Positive FHR at 6 weeks	0.08 [0.01; 0.50]	0.025	0.72 [0.03; 19.6]	0.82

Data show coefficients (β) ± standard error for reproductive variables and OR [95% CI] for pregnancy outcomes. Models were adjusted by age, smoking habits, race, body mass index, and neutrophil levels considering the control as a reference group. The *p*-value was extracted from the significance of each factor. Ovarian endometriosis (OMA); surgery (S); anti-Müllerian hormone (AMH); β-chorionic hormone (βCH); fetal heart rate (FHR).

**Table 5 biomedicines-11-00844-t005:** Associations of endometriosis with reproductive and pregnancy outcomes.

**Reproductive Variables**	**OMA + S**	** *p* ** **-Value**
AMH (ng/mL)	−0.66 ± 0.28	0.020
Follicular count	−0.27 ± 0.85	0.75
Number of cycles	−0.01 ± 0.14	0.93
Oocytes/cycle	−1.40 ± 1.57	0.37
Embryos C + D/cycle	0.30 ± 0.72	0.68
**Pregnancy outcomes**	**OMA + S**	** *p* ** **-Value**
Biochemical pregnancy (positive βCH)	0.98 [0.39; 2.46]	0.97
Positive FHR at 6 weeks	0.08 [0.001; 0.50]	0.06

Data show coefficients (β) ± standard error for reproductive variables and OR [95% CI] for pregnancy outcomes. Models were adjusted by age, smoking habits, race, body mass index, and neutrophil levels considering the control as a reference group. The *p*-value was extracted from the significance of each factor. Ovarian endometriosis (OMA); surgery (S); anti-Müllerian hormone (AMH); β-chorionic hormone (βCH); fetal heart rate (FHR).

**Table 6 biomedicines-11-00844-t006:** Summary of the effectiveness of different treatments for women with endometriosis and in vitro fertilization (IVF) and reproductive and pregnancy outcomes.

Author, Year	Study Design	Women	Comparison Treatment before IVF	Results in IVF
Wu et al., 2019 [36]	Metanalysis	2878	Expectant management versus surgical treatment	No differences in pregnancy rate, miscarriage, and mature oocytes retrieved. However, a low number of oocytes was retrieved in the surgical group
Tao et al., 2017 [37]	Metanalysis	2649	Surgical versus non-surgical treatment	No differences in the duration of IVF, formed embryos, pregnancy rates, and live birth rates
Hamdam et al., 2015 [21]	Metanalysis	2977	Surgical versus non-surgical treatment	No differences in clinical pregnancy rate, live birth rate, and number of oocytes retrieved
Nickkho-Amiry et al., 2018 [38]	Metanalysis	-	Surgical versus non-surgical treatment	No differences in number of oocytes retrieved, pregnancy, and live birth rates
Opøien et al., 2011 [39]	Observational	661	Exploratory laparoscopy versus surgical removal of lesions	Improved implantation rate, pregnancy rate, and live birth rate in the group with complete excision of endometriosis lesions
Frangež et al., 2022 [40]	Observational	436	Cystectomy of endometriomas versus women with other infertility issues and without endometriosis	Differences in gonadotropins controlling ovarian stimulation (high doses in cystectomy group) and number of oocytes retrieved (low in endometriosis group). No differences in pregnancy rates

## Data Availability

Data will be sent upon request to the corresponding author after evaluating the scientific/academic interest of the proposal.

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
