# Peer review of "Impact of Ovarian Endometrioma and Surgery on Reproductive Outcomes: A Single-Center Spanish Cohort Study"

_biomedicines, 2023, doi:10.3390/biomedicines11030844_

Round 1

Reviewer 1 Report

Interesting paper. Indeed, as stressed in the paper, reproductive variables in IVF treated women with and without endometriosis studies are used to identify differences garding our data in women with IVF, low rate of reproductive variables AMH, follicle  count, number of oocytes and number of embryos. Therefore, this paper has a good portion of novelty for searching biomarkers. 

I have some suggestions to improve this paper:

1.       It would be worthwhile to explain what therapeutic options are currently available to alleviate the symptoms of endometriosis and to prevent its development.

2.       In addition, more information on the applicability and side effects of currently used protocol would be needed to emphasize the possible better detection and the importance of developing appropriate therapy. (It could even be summarized in a table.)

3.       The quality of the manuscript would be improved if some experimental results were included for the critical point of view of different used methods. In this way, readers can better see how effective/well-applied the different test used to see how endometriosis decreases the ovarian reserve. (It could even be summarized in a table.)

4. In general, I recommend authors use more references and to add more material in discussion section to back their claims. Nonetheless, in my opinion, less than 50-60 articles for a paper are insufficient. I believe that adding more citations will help to provide better and more accurate background to this study. 

Author Response

Response: Thank you very much for your kind words and for taking the time to review our article. We answer point-by-point your queries.

  1. It would be worthwhile to explain what therapeutic options are currently available to alleviate the symptoms of endometriosis and to prevent its development.

Response: This aspect has been expanded in the introduction (lines 46-54).

  1. In addition, more information on the applicability and side effects of currently used protocol would be needed to emphasize the possible better detection and the importance of developing appropriate therapy (It could even be summarized in a table).

Response: This is a great point which was described in the discussion (lines 219-226).

  1. The quality of the manuscript would be improved if some experimental results were included for the critical point of view of different used methods. In this way, readers can better see how effective/well-applied the different test used to see how endometriosis decreases the ovarian reserve (It could even be summarized in a table).

Response: A summary of observational studies and meta-analyses has been included in Table 6 with this aspect.            

  1. In general, I recommend authors use more references and to add more material in discussion section to back their claims. Nonetheless, in my opinion, less than 50-60 articles for a paper are insufficient. I believe that adding more citations will help to provide better and more accurate background to this study.

Response: We fully agree. We have updated the introduction and discussion and extended the list of references (going from 36 to 48).

Reviewer 2 Report

This is a retrospective observational study of women with and without ovarian endometriosis undergoing IVF.

The article is relevant to the journal and the study methodology is good. 

English grammar and language is appropriate and well understood

Minor corrections: 

LINE 13-14 Overclutter sentence. Please rephrase :  "The literature has promoted a reduction of Anti-Müllerian hormone (AMH) and antral follicular count (AFC) in women with ovarian endometrioma (OMA), undergoing in-vitro fertilization (IVF), being further declined after surgery" 

underwent --> undergoing

LINE 66-70 Please rephrase the following. : "In the present study, it was planted as primary aim to compare the reproductive variables in women under IVF with and without endometriosis. In addition, as a secondary aim, we consider exploring if the reproductive variables were modified by ovarian endometrioma surgery.  "  Please use outcome instead of aim

line 266 & 324 Please correct quistectomy to cystectomy

Author Response

Response: Thank you for your kind words and time to review our article. We have corrected your queries.

Minor corrections:

  • LINE 13-14 Overclutter sentence. Please rephrase: "The literature has promoted a reduction of Anti-Müllerian hormone (AMH) and antral follicular count (AFC) in women with ovarian endometrioma (OMA), undergoing in-vitro fertilization (IVF), being further declined after surgery"
  • underwent --> undergoing
  • LINE 66-70 Please rephrase the following: "In the present study, it was planted as primary aim to compare the reproductive variables in women under IVF with and without endometriosis. In addition, as a secondary aim, we consider exploring if the reproductive variables were modified by ovarian endometrioma surgery. “Please use outcome instead of aim
  • line 266 & 324 Please correct quistectomy to cystectomy.

Response: Thank you for these suggestions. The text has been revised and updated.